# The Roles of NFR2-Regulated Oxidative Stress and Mitochondrial Quality Control in Chronic Liver Diseases

**DOI:** 10.3390/antiox12111928

**Published:** 2023-10-29

**Authors:** Jeong-Su Park, Nodir Rustamov, Yoon-Seok Roh

**Affiliations:** College of Pharmacy and Medical Research Center, Chungbuk National University, Cheongju 28160, Republic of Korea; 6318js@cbnu.ac.kr (J.-S.P.); 2022230058@cbnu.ac.kr (N.R.)

**Keywords:** oxidative stress, CLD, antioxidants, ROS, mitochondria, NRF2

## Abstract

Chronic liver disease (CLD) affects a significant portion of the global population, leading to a substantial number of deaths each year. Distinct forms like non-alcoholic fatty liver disease (NAFLD) and alcoholic fatty liver disease (ALD), though they have different etiologies, highlight shared pathologies rooted in oxidative stress. Central to liver metabolism, mitochondria are essential for ATP production, gluconeogenesis, fatty acid oxidation, and heme synthesis. However, in diseases like NAFLD, ALD, and liver fibrosis, mitochondrial function is compromised by inflammatory cytokines, hepatotoxins, and metabolic irregularities. This dysfunction, especially electron leakage, exacerbates the production of reactive oxygen species (ROS), augmenting liver damage. Amidst this, nuclear factor erythroid 2-related factor 2 (NRF2) emerges as a cellular protector. It not only counters oxidative stress by regulating antioxidant genes but also maintains mitochondrial health by overseeing autophagy and biogenesis. The synergy between NRF2 modulation and mitochondrial function introduces new therapeutic potentials for CLD, focusing on preserving mitochondrial integrity against oxidative threats. This review delves into the intricate role of oxidative stress in CLD, shedding light on innovative strategies for its prevention and treatment, especially through the modulation of the NRF2 and mitochondrial pathways.

## 1. Introduction

Chronic liver diseases (CLDs), which encompass a range of conditions, are a major health burden affecting millions of people worldwide. These include NAFLD, nonalcoholic steatohepatitis (NASH), ALD, cirrhosis, and hepatocellular carcinoma (HCC) [1]. These diseases share the common features of progressive ROS production, oxidative stress, liver deterioration, inflammation, and fat accumulation within the liver, ultimately leading to compromised liver function and potential long-term complications [2,3]. The impact of CLDs on public health is substantial, with an estimated 1.5 billion people affected globally [4]. Tragically, these diseases contribute to approximately 2 million deaths each year [4]. Within the realm of CLD, NAFLD stands out because of its alarming prevalence and strong correlation with conditions such as metabolic syndrome, oxidative stress, mitochondrial dysfunction, and obesity [5,6]. Approximately 25% of the global population is affected by NAFLD [7].

These diseases are characterized by their intricate pathogeneses, highlighting the need for a comprehensive understanding of the mechanisms underlying their pathophysiologies. Interestingly, their progression provides crucial insights into the development of targeted interventions. The development of NAFLD is initiated via the accumulation of excess fat within hepatocytes, a state known as simple steatosis [8]; this accumulation results from an imbalance between the influx and clearance of triglycerides [9]. Insulin resistance and increased lipogenesis contribute to enhanced lipid accumulation [10]. Dysfunctional adipose tissue leads to an increased release of free fatty acids, contributing to hepatic fat accumulation [11]. As simple steatosis progresses to NASH, cellular stress and inflammation intensify [12]. Concurrently, adipokines and pro-inflammatory cytokines drive inflammation and oxidative stress, further disrupting liver homeostasis.

In contrast, ALD is primarily caused by chronic excessive alcohol consumption, resulting in a continuum of liver conditions that may evolve into cirrhosis and HCC [13]. The initial stage of ALD is liver steatosis driven by alcohol-induced metabolic changes. Hepatocytes metabolize ethanol through various enzymatic pathways, and alcohol dehydrogenase (ADH) and cytochrome P450 2E1 (CYP2E1) convert ethanol into acetaldehyde, a highly toxic substance that damages liver cells and impairs DNA [14]. Moreover, ethanol metabolism generates ROS, triggering oxidative stress that leads to cellular damage [15], protein and DNA modifications [16], and lipid peroxidation [17]. 

Although NAFLD and ALD have distinct triggers, they share pivotal mechanisms that accelerate liver disease [18]. Both these conditions involve hepatocellular injury and inflammation. Inflammatory cytokines contribute to the progression of both NAFLD and ALD by fostering a pro-inflammatory microenvironment that fuels disease progression. Cellular stress is amplified as NAFLD and ALD progress, setting the stage for subsequent stages. Moreover, the progression of NAFLD and ALD is associated with an increased risk of cirrhosis [19]. Cirrhosis is the end-stage manifestation of both conditions. Continued inflammation, oxidative stress, and fibrosis drive the transformation of healthy liver tissue into fibrous scar tissue, which compromises liver function and culminates in cirrhosis [20,21]. Cirrhosis, often a result of long-standing NAFLD or ALD, creates an environment conducive to the development of HCC. Chronic inflammation and cellular stress drive genetic and epigenetic changes, promoting the transition of hepatocytes toward malignant growth [22].

While CLDs have distinct etiologies, they share common mechanistic pathways, particularly the pivotal role of oxidative stress [15,23,24]. The imbalance between the production of ROS and antioxidant defense serves as a unifying factor that accelerates the progression of CLD. Oxidative-stress-induced cellular damage, lipid peroxidation, and mitochondrial dysfunction contribute to inflammation, cellular stress, and hepatocellular injury, ultimately leading to cirrhosis and the development of HCC [25,26,27]. In this review, we summarize the current knowledge about the molecular mechanisms underlying oxidative stress in CLDs and identify innovative approaches to prevent, manage, and treat them.

## 2. Oxidative Stress and Antioxidant Defense Mechanisms in CLDs

### 2.1. Definition of Oxidative Stress

Oxidative stress is a crucial concept in understanding cellular and systemic health. At its core, it denotes a situation wherein there is a significant imbalance in cell antioxidative homeostasis. This imbalance does not simply emerge spontaneously; it is often the consequence of either a surge in the production of ROS or a decline in the body’s capacity to neutralize these reactive molecules effectively [3,28,29,30]. What makes ROS particularly intriguing, yet hazardous, is their high reactivity. The term “reactive oxygen species” encompasses a range of molecules, including free radicals like superoxide anions (O_2_^•−^) and hydroxyl radicals (•OH), as well as non-radical species like hydrogen peroxide (H_2_O_2_) [31]. Each of these molecules has a propensity to engage in chemical reactions with various cellular components, potentially disrupting their normal function. For instance, they can damage DNA, proteins, and lipids, all of which are essential for cell function and integrity.

### 2.2. The Formation of Reactive Oxygen Species

Free radicals and reactive oxygen species are inherent outcomes of various biological mechanisms. They emerge as natural byproducts of cellular metabolism, chiefly arising from processes like mitochondrial respiration, immune responses, and enzymatic reactions [32].

For instance, during oxidative phosphorylation, a process in which cells produce energy in the mitochondria, electrons sometimes leak from the electron transport chain, resulting in the incomplete reduction of molecular oxygen (O_2_) to produce the superoxide anion (O_2_^•−^), a primary ROS [33]. The rate and extent of electron leakage can be influenced by various factors, including mitochondrial membrane potential, ambient oxygen concentration, and the presence of certain electron donors or acceptors [34,35]. A more detailed role of mitochondria in the formation of ROS will be discussed later in the review.

In addition to these sources, some specific enzymes play a role in the production of ROS. Enzymes such as NADPH oxidases, xanthine oxidase, and cytochrome P450 often yield ROS as byproducts during typical metabolic reactions [36]. These enzymes facilitate electron transfers to molecular oxygen, subsequently producing superoxide anions (O_2_^•−^) or hydrogen peroxide (H_2_O_2_) [36]. Moreover, during inflammation or when the immune system is active against pathogens, certain phagocytic cells, like neutrophils and macrophages, ramp up the production of ROS. This is primarily a defense strategy wherein these cells activate NADPH oxidase to produce vast quantities of superoxide anions (O_2_^•−^) aimed at neutralizing or killing the invading microorganisms [37]. Further expanding on cellular metabolism, ROS are also products of the metabolic breakdown of various compounds. Molecules such as fatty acids, amino acids, and sugars, when metabolized, often yield ROS like hydrogen peroxide (H_2_O_2_). These reactions are frequently catalyzed by a range of enzymes, including specific oxidases [38].

### 2.3. Antioxidant Defense Mechanism

It is important to note that while free radicals and ROS are normal byproducts of cellular metabolism, their excessive and uncontrolled production can lead to oxidative stress, which can damage cellular components by oxidizing lipids, disrupting protein structure, and causing DNA strand breaks [39], thereby contributing to impaired cellular function, tissue inflammation, and the development or progression of CLD [40]. The body has natural defense mechanisms, including antioxidant enzymes and non-enzymatic antioxidants, to neutralize and detoxify ROS and maintain the redox balance. 

However, oxidative stress occurs when the production of ROS overwhelms the capacity of antioxidants to neutralize them. Antioxidants, both enzymatic and non-enzymatic, scavenge ROS and prevent oxidative damage to biomolecules, including lipids, proteins, and DNA [41].

#### 2.3.1. Enzymatic Antioxidants

Enzymatic antioxidants comprise a set of enzymes vital for counteracting ROS and shielding cells from the detrimental effects of oxidative stress. These enzymes aid in the transformation of harmful ROS into molecules that are either less reactive or entirely benign.

Among them, superoxide dismutase (SOD) serves as one of the cornerstones of this protective mechanism. This enzyme specifically targets superoxide anions (O_2_^•−^) converting them into the less harmful hydrogen peroxide (H_2_O_2_) and benign molecular oxygen (O_2_) [42]. It presents itself in three diverse isoforms, copper–zinc SOD (CuZnSOD), manganese SOD (MnSOD), and extracellular SOD (EcSOD). Each isoform has evolved to function optimally in different cellular environments, ensuring that ROS are managed efficiently across various cellular compartments [43].

Another pivotal enzyme is catalase, which is primarily found in peroxisomes. Its chief function involves decomposing hydrogen peroxide (H_2_O_2_) into water (H_2_O) and molecular oxygen (O_2_) [44]. By doing so, catalase acts as a defense mechanism, deterring the buildup of H_2_O_2_ which, if left unchecked, could result in the creation of the immensely reactive hydroxyl radicals (^•^OH), especially under the conditions of CLD [45,46].

The glutathione peroxidase (GPx) family of enzymes also assumes a prominent role, particularly in the context of chronic liver diseases. Utilizing stores of reduced glutathione (GSH), these enzymes neutralize the threats posed by hydrogen peroxide and organic hydroperoxides, transforming them into corresponding alcohols [47]. Each GPx isoform displays unique cellular localization and substrate affinities, positioning them aptly to counter distinct oxidative threats in specific tissues or cellular regions [48].

Furthermore, glutathione reductase is instrumental in sustaining the reduced state of GSH, an indispensable cellular antioxidant [49]. This enzyme is responsible for reviving GSH from its oxidized variant (GSSG), guaranteeing a steady reservoir of GSH to support diverse antioxidant activities [49].

Lastly, thioredoxin reductase is vital for re-establishing the reduced form of thioredoxin, playing an integral part in upholding the cellular redox equilibrium [50]. By restoring reduced thioredoxin, this enzyme fortifies cellular defenses against oxidative duress and bolsters a range of redox-reliant processes [51].

#### 2.3.2. Non-Enzymatic Antioxidants

Non-enzymatic antioxidants are another group of molecules that help neutralize ROS and protect cells from oxidative damage. Unlike enzymatic antioxidants, these molecules do not rely on specific enzymes to exert their protective effects. Instead, they directly scavenge free radicals and regenerate other antioxidant molecules, safeguarding the cell’s integrity and function.

GSH, a tripeptide comprising cysteine, glutamic acid, and glycine, stands out as a predominant intracellular antioxidant [52]. It is one of the most abundant antioxidants in cells and plays a crucial role in maintaining cellular redox balance. GSH acts as a direct scavenger of ROS, particularly hydrogen peroxide (H_2_O_2_), and can regenerate other antioxidants, such as vitamins C and E [52].

Vitamin C, also known as ascorbic acid, is a water-soluble antioxidant crucial for neutralizing ROS and protecting against oxidative damage. It donates electrons, regenerates vitamin E, and enhances the antioxidant capacity of other compounds [53]. On the other hand, vitamin E, which encompasses both tocopherols and tocotrienols, is fat-soluble and intercepts lipid-based free radicals, thus preventing lipid peroxidation in cellular membranes [54]. Vitamin E also exhibits anti-inflammatory properties and protects the liver against oxidative-stress-induced damage [55].

Another group of non-enzymatic antioxidants, carotenoids, are pigment molecules which give many fruits and vegetables their vibrant colors and include well-known variants such as beta-carotene, lycopene, lutein, and zeaxanthin [56]. Carotenoids possess antioxidant properties and scavenge ROS, particularly singlet oxygen [56]. In the context of CLD, they are instrumental in shielding cells, tissues, and organs from oxidative harm [57,58].

Flavonoids are a diverse group of polyphenolic compounds found in fruits, vegetables, teas, and cocoa [59]. They possess antioxidant and anti-inflammatory properties, and their chemical structure allows them to scavenge several types of free radicals. Beyond their role as free radical scavengers, flavonoids can also modulate cellular signaling pathways, regulate enzyme activity, and influence gene expression [60]. Recognized for their potential health advantages, flavonoids have been linked to protective effects against oxidative stress-related ailments, including CLD. 

Lastly, alpha-lipoic acid is a sulfur-containing antioxidant that functions in both water- and lipid-based environments [61]. It acts as a free radical scavenger and can regenerate other antioxidants such as vitamins C and E. Alpha-lipoic acid also plays a role in cellular energy metabolism and has been investigated for its potential therapeutic effects in CLD [62,63].

### 2.4. Oxidative Stress Biomarkers in CLD

The evaluation of oxidative stress in CLD is important for understanding disease progression, monitoring treatment response, predicting prognosis, identifying therapeutic targets, and providing individualized care. However, measuring oxidative stress in the body is challenging owing to the transient nature and rapid reactivity of ROS. Biomarkers of oxidative stress, such as lipid peroxidation products, protein carbonyls, and DNA oxidation products, can be used to assess the extent of oxidative damage. 

Among many biomarkers, malondialdehyde (MDA) stands out. This reactive aldehyde, produced via the peroxidation of polyunsaturated fatty acids, is a key marker of lipid peroxidation [64]. Recognized as a biomarker of lipid peroxidation, MDA serves as a prominent indicator of oxidative stress. Elevated MDA levels signify heightened oxidative damage to cellular lipids and membranes [64]. Following the trail of oxidative damage, 8-Hydroxy-2′-deoxyguanosine (8-OHdG) emerges as another significant marker. Originating from DNA oxidation, 8-OHdG acts as an indicator of oxidative DNA damage [65]. Increased levels of 8-OHdG in urine, blood, and liver tissues indicate DNA damage caused by ROS [65,66].

Carbonylation occurs when proteins are oxidatively modified, resulting in the formation of carbonyl groups [67]. The protein carbonyl content serves as an indicator of oxidative damage to proteins. Elevated levels of protein carbonyls reflect the oxidative stress burden on proteins in CLDs [68].

On a cellular level, glutathione, a vital intracellular antioxidant, plays a pivotal role in neutralizing ROS [69]. Monitoring the balance between reduced GSH and oxidized glutathione disulfide (GSSG) can provide insights into the redox status of cells. A dwindling GSH/GSSG ratio is a warning of intensifying oxidative stress and an ailing antioxidant defense in CLD [70].

The activity of SOD serves as an indirect barometer of antioxidant capacity against superoxide radicals. In the realm of CLD, diminishing SOD activity suggests an eroding defense mechanism against oxidative onslaughts [71].

Moreover, the total antioxidant capacity (TAC) offers a broader perspective. Representing the cumulative antioxidant power of biological samples, including plasma and liver tissues, the TAC evaluates the efficacy of both enzymatic and non-enzymatic antioxidants at quelling free radicals. A declining TAC level underscores a mounting discord between pro-oxidants and antioxidants [72].

Lastly, advanced oxidation protein products (AOPPs) further highlight the intricacies of oxidative stress. Emerging from protein oxidation, AOPPs lead to the birth of advanced glycation end products (AGEs) [73]. AOPP levels increase in response to oxidative stress and can serve as markers of protein damage and inflammation in CLDs [74].

The interpretation and clinical utility of oxidative stress biomarkers should be considered in the contexts of specific liver diseases. These biomarkers can provide valuable insights into the oxidative stress burden, disease progression, and potential therapeutic interventions targeting oxidative stress in CLDs. However, a comprehensive evaluation of multiple biomarkers is often necessary to accurately assess the overall oxidative stress status.

## 3. Oxidative Stress and NRF2 Signaling Pathways in Chronic Liver Disease

### 3.1. The NRF2-KEAP1 Pathway as a Key Regulator of Oxidative Stress in Liver Health

NRF2 is a critical transcription factor that plays a pivotal role in cellular defense against oxidative stress resulting from a high level of ROS [75]. NRF2 is involved in various cellular functions, including detoxification and the regulation of cell metabolism [76]. It has particular relevance considering that oxidative stress is a leading cause of liver disease [77].

Under normal physiological conditions, NRF2 is tightly regulated by its interaction with Kelch-like ECH-associated protein 1 (KEAP1), a primary inhibitor of NRF2. KEAP1 acts as a sensor for redox reactions, and together with CULLIN3, forms an E3 ubiquitin ligase complex responsible for targeting NRF2 for ubiquitination and subsequent degradation. This tight regulation ensures that NRF2 activity is precisely controlled, preventing the uncontrolled activation of cytoprotective genes [78] (Figure 1).

Notably, critical cysteine residues on KEAP1 are modified in response to oxidative stress or increased ROS, leading to the disruption of the KEAP1–NRF2 complex; additionally, certain chemicals, known as NRF2 activators, can also modify this complex, illustrating the diverse regulatory mechanisms in the KEAP1–NRF2 pathway [79,80]. Consequently, NRF2 is stabilized and accumulates within cells. The accumulated NRF2 translocates to the nucleus, where it forms a heterodimer with a small musculoaponeurotic fibrosarcoma oncogene homolog (sMAF). The NRF2–sMAF complex binds to antioxidant response elements (AREs) within the promoter regions of target genes, initiating their transcription. The transcriptional program orchestrated by NRF2 activation encompasses a broad range of cytoprotective genes, including those encoding phase II detoxification enzymes, such as NAD(P)H quinone oxidoreductase 1 (NQO1), glutathione S-transferases (GSTs), and heme oxygenase-1 (HO-1), which play crucial roles in neutralizing toxic compounds and reactive metabolites [81]. This regulatory network encompasses antioxidant metabolism, lipid metabolism, protein degradation, and inflammation regulation and reduces the activation of hepatic stellate cells (HSCs), thereby ensuring the maintenance of cellular homeostasis and resilience against diverse stressors and thus contributing to liver health and protection against oxidative damage [82].

### 3.2. The NRF2-Mediated Regulation of Lipid Metabolism in CLDs

In addition to NRF2’s primary function in regulating antioxidant pathways, it has also been shown to impact lipid metabolism, making it a topic of growing interest, especially in the context of CLDs [83,84].

The dysregulation of lipid metabolism plays a significant role in CLDs such as NAFLD and NASH. The activation of NRF2 because of oxidative stress or corresponding activators has emerged as a promising therapeutic approach to addressing lipid metabolism abnormalities and mitigating disease progression in chronic liver conditions [85]. It exerts its effects on lipid metabolism via several interconnected mechanisms. A notable study emphasized Nrf2’s regulatory influence on hepatic lipid accumulation induced via a high-fat diet. It was observed that an Nrf2 deficiency augments lipogenesis primarily through enhancing the activity of sterol regulatory element-binding protein-1c (SREBP-1c), a fundamental molecule in lipid synthesis. Moreover, the absence of Nrf2 diminishes autophagic flux and hinders the fusion of autophagosomes with lysosomes, causing a reduction in lipolysis in the liver and leading to lipid accumulation [84]. Conversely, when examining the effects of a methionine- and choline-deficient diet, which typically induces fatty liver, it was found that amplifying Nrf2 expression in mice counteracted the fatty liver condition, suggesting Nrf2’s protective or remedial role against fatty liver disease under particular dietary circumstances. Additionally, in high-fat-diet-resultant liver diseases, NRF2 activation prevented the adverse effects of the diet, including increased body weight, adipose mass, and hepatic lipid accumulation in wild-type mice. In cells undergoing adipogenesis, the activation of NRF2 inhibited lipid accumulation, and its influence led to a marked downregulation of the genes responsible for fatty acid synthesis in the liver [86].

Another significant effect of NRF2 activation is the promotion of β-oxidation, the process through which fatty acids are broken down in the mitochondria to produce energy [87]. NRF2 induces the expression of genes associated with fatty acid transport, such as carnitine palmitoyl transferase 1A (*CPT1A*) and acyl-CoA dehydrogenases (*ACADs*), thereby facilitating the transport of fatty acids into the mitochondria for β-oxidation [88]. Furthermore, NRF2 enhances the expression of key enzymes in the β-oxidation pathway, such as carnitine acylcarnitine translocase (CACT), which results in an increased breakdown of triglycerides into fatty acids and glycerol. This heightened β-oxidation helps metabolize accumulated lipids and reduces the hepatic lipid content, thus ameliorating hepatic steatosis [89]. Moreover, NRF2 activation plays a crucial role in reducing oxidative stress and is strongly associated with the pathogeneses of CLDs. Oxidative stress leads to lipid peroxidation, which causes cellular damage and impairs lipid metabolism [90]. By mitigating oxidative stress, NRF2 protects hepatocytes from lipid-induced injury and maintains lipid homeostasis.

### 3.3. NRF2-Mediated Protection against Lipid Peroxidation

NRF2 plays a crucial role in regulating lipid peroxidation and protecting cells from its detrimental effects. Lipid peroxidation, a result of ROS action and oxidative stress, plays a pivotal role in driving pathological processes in the liver and contributes to inflammation, fibrosis, and cellular damage [91]. The excessive accumulation of ROS within cells initiates a series of chemical reactions that lead to the oxidation of the polyunsaturated fatty acids (PUFAs) present in lipid molecules. PUFAs with multiple double bonds are particularly susceptible to oxidation [92]. During lipid peroxidation, ROS, such as the hydroxyl radical (OH^•^), abstract a hydrogen atom from neighboring PUFAs in lipid molecules, leading to the formation of highly reactive lipid radicals (L^•^) and water (H_2_O). The lipid radicals (L^•^) then react with molecular oxygen (O_2_) to generate lipid peroxyl radicals (LOO^•^). This initiates a chain reaction that propagates lipid peroxidation throughout the cellular membranes [93]. As lipid peroxidation progresses, reactive aldehydes, such as MDA and 4-hydroxynonenal (4-HNE), are produced [64]. These highly reactive aldehydes interact with cellular macromolecules, including nucleic acids [94], and other lipids [95]. The accumulation of lipid peroxidation products intensifies cellular damage and perpetuates oxidative stress, further contributing to the onset of CLDs such as NAFLD, NASH, and ALD [95,96,97]. However, amidst this oxidative onslaught, NRF2 strengthens cellular defenses, particularly by upregulating enzymes such as glutathione S-transferases (GSTs). These crucial enzymes attach GSH to the products of lipid peroxidation, mitigating their damaging effects and emphasizing NRF2’s pivotal role in cellular protection [98] (Figure 1).

In the absence of adequate NRF2 response, lipid peroxidation can induce inflammation via the NF-κB pathway [99], a key regulator of inflammation. Reactive aldehydes from lipid peroxidation can activate NF-κB, a central regulator of inflammation [100]. Under such circumstances, NRF2 serves as a counteractive force. NRF2 not only upregulates the expression of antioxidant enzymes like HO-1 and NQO1 but also plays a role in maintaining redox homeostasis through the regulation of glutathione synthesis and recycling, further helping to detoxify reactive aldehydes and other byproducts of oxidative stress [101]. This robust antioxidant response orchestrated by NRF2 effectively reduces the pro-inflammatory response instigated by NF-κB [83,102].

Continued inflammation from lipid-peroxidation-derived aldehydes significantly contributes to the activation of HSCs, a defining event in fibrosis [103]. Inflammatory signals within the liver microenvironment stimulate the transition of quiescent HSCs into an activated myofibroblast-like phenotype. Activated HSCs become the main source of excess ECM proteins like collagen and fibronectin, leading to ECM accumulation and remodeling. This disrupts the liver’s structure and impairs its function. These HSCs also intensify inflammation by producing proinflammatory substances [104]. The ongoing inflammation and fibrogenesis form a cycle that exacerbates liver damage and fibrosis. Persistent liver injury can advance fibrosis to cirrhosis, further affecting liver function and raising the risk of complications like HCC [105]. NRF2’s protective role in this scenario is pivotal. Apart from promoting the elimination of reactive lipid derivatives, NRF2 activation also downregulates profibrogenic genes in HSCs, thereby reducing their activation potential [106]. Together, these NRF2-mediated responses not only curb the immediate threats posed by lipid peroxidation but also play a more long-term role in safeguarding liver health and preventing fibrotic changes [107].

### 3.4. The Interplay of NRF2 and NF-κB in the Modulation of Inflammation via Kupffer Cells

In chronic liver disease, Kupffer cells, the resident macrophages of the liver, play a pivotal role in inflammation. The activation and subsequent inflammation of Kupffer cells can be initiated via various stimuli, such as hepatotoxic agents, damaged liver cells, or pathogens [108]. Within a Kupffer cell, NRF2 acts as a key modulator of the inflammatory response and contributes to cellular homeostasis and tissue integrity in mitigating liver inflammation [109,110].

NRF2 regulates inflammation through multiple mechanisms. It promotes the expression of genes encoding antioxidant enzymes, including HO-1, SOD, and GPx. These enzymes play crucial roles in neutralizing ROS and reducing oxidative stress, which are known to trigger inflammation [111,112]. Additionally, NRF2 promotes the production of anti-inflammatory mediators that help suppress inflammatory responses. One such mediator is interleukin-10 (IL-10), an immunomodulatory cytokine that inhibits the production of pro-inflammatory cytokines. NRF2 stimulates the expression of IL-10 by binding to AREs in the IL-10 promoter. IL-10 acts as a negative feedback regulator that attenuates the inflammatory cascade and promotes tissue resolution [113,114].

In Kupffer cells, NRF2 and NF-κB represent two pivotal transcription factors that play contrasting roles in the regulation of inflammation. NRF2 curtails the inflammatory effects of NF-κB by impeding its signaling pathway [115,116]. NRF2 interferes with NF-κB activation through multiple mechanisms. It also competes with NF-κB for coactivators required for gene transcription that enhance the activity of transcription factors by facilitating their interactions with the transcriptional machinery. By effectively competing with NF-κB for these coactivators, NRF2 limits their availability, resulting in reduced NF-κB transcriptional activity and dampening the NF-κB-dependent expression of pro-inflammatory genes [117,118,119]. Thus, NRF2 attenuates the pro-inflammatory response by reducing NF-κB-mediated gene expression.

Another mechanism through which NRF2 inhibits NF-κB signaling involves the regulation of IκB (inhibitor of NF-κB) proteins. IκB proteins play a crucial role in controlling NF-κB activation. They sequester NF-κB in the cytoplasm, thereby preventing its translocation to the nucleus and inhibiting its transcription [115,120]. NRF2 activation induces IκB proteins, including IκBα, leading to the enhanced retention of NF-κB in the cytoplasm and restricting its nuclear translocation, thereby reducing pro-inflammatory signaling [120] (Figure 2).

Furthermore, the degradation of IκB proteins is tightly regulated and critical for NF-κB activation [121]. Cellular exposure to pro-inflammatory stimuli causes IκB phosphorylation, followed by ubiquitination and proteasomal degradation. This frees NF-κB, allowing for its nuclear translocation, where it activates the transcription of pro-inflammatory mediators. However, NRF2 inhibits the phosphorylation and subsequent degradation of IκB proteins. By preventing IκB degradation, NRF2 aids the cytoplasmic sequestration of NF-κB, restricting its activity and reducing its pro-inflammatory signaling [122].

Through these mechanisms, NRF2 effectively modulates NF-κB signaling and the subsequent expression of pro-inflammatory genes. This crosstalk between NRF2 and NF-κB contributes to the fine-tuned regulation of inflammation by NRF2.

### 3.5. The Role of NRF2 in Mitigating Oxidative-Stress-Induced HSC Activation

NRF2 activation plays a crucial role in ameliorating liver fibrosis by interfering with the activation of HSCs, which are the key drivers of excessive ECM production. NRF2 exerts its inhibitory effects through multiple interconnected mechanisms [123]. One of the major mechanisms though which NRF2 inhibits HSC activation is the modulation of the transforming growth factor-beta (TGF-β) signaling pathway [124]. TGF-β is a potent profibrogenic cytokine that promotes HSC activation and ECM synthesis. NRF2 activation disrupts the Smad signaling pathway downstream of TGF-β, preventing the formation of Smad complexes and their translocation to the nucleus [125]. Consequently, the expression of fibrotic genes is suppressed, leading to reduced HSC activation and ECM production. In addition to interfering with TGF-β signaling, NRF2 activation modulates the expression of various fibrogenic mediators involved in HSC activation. For example, NRF2 downregulates the production of platelet-derived growth factor, connective tissue growth factor, and endothelin-1, which are potent stimulators of HSC activation and ECM synthesis. Thus, NRF2 limits HSC activation and attenuates fibrosis progression by inhibiting the expression of these fibrogenic mediators [126] (Figure 2).

Moreover, NRF2 activation exerts anti-inflammatory effects on HSCs, which are crucial for the development of fibrosis. It inhibits the release of pro-inflammatory cytokines and chemokines such as interleukin-6 (IL-6) and tumor necrosis factor-alpha (TNF-α), thereby reducing the inflammatory milieu that promotes HSC activation [127]. The anti-inflammatory effect of NRF2 attenuates fibrogenesis and limits ECM production in HSCs [128]. Additionally, NRF2 activation reduces oxidative stress in HSCs, which is another key driver of the activation and progression of fibrosis. By enhancing the cellular antioxidant defense system, NRF2 upregulates the expression of antioxidant enzymes that scavenge ROS, thereby reducing oxidative damage in HSCs [129,130]. Reduced oxidative stress inhibits HSC activation and subsequent ECM production. Furthermore, NRF2 activation modulates epigenetic mechanisms in HSCs. It influences DNA methylation patterns and histone modifications, which can lead to the suppression of fibrotic gene expression in HSCs [131].

In summary, NRF2 activation ameliorates liver fibrosis by interfering with HSC activation through modulating TGF-β signaling, downregulating fibrogenic mediators, suppressing inflammation, reducing oxidative stress, and epigenetic regulation. These interconnected mechanisms work together to limit HSC activation, ECM production, and fibrosis progression in CLDs.

## 4. Crosstalk between NRF2 and Mitochondria Quality Control in Chronic Liver Disease

### 4.1. The Role of Mitochondria in the Formation of ROS

Mitochondria, often referred to as the powerhouses of cells, play a pivotal role in producing ROS and driving oxidative stress, especially in the context of chronic liver disease [132]. One primary source of ROS in mitochondria is electron leakage from the electron transport chain (ETC), which is fundamental in generating cellular energy through oxidative phosphorylation. While this chain mostly transfers electrons efficiently to molecular oxygen, a small fraction can escape prematurely, resulting in the creation of superoxide anion (O_2_^•−^) as a primary ROS. Components such as complex I and complex III of the ETC are recognized as significant sites for the production of ROS. Electron leakage, along with inefficient electron transfer and compromised mitochondrial membrane potential, can escalate the output of ROS [133]. Several factors can induce these dysfunctions, including mitochondrial DNA mutations, calcium accumulation, the opening of the mitochondrial permeability transition pore (mPTP), oxidative damage to ETC components diminishing the availability of electron carriers, and issues with protein assembly [34,134]. Mitochondrial DNA (mtDNA), which is situated close to ROS production locations, is exceptionally vulnerable to oxidative harm. Damaged mtDNA can further debilitate mitochondrial functionality, promoting the generation of ROS and establishing a cycle of increasing mitochondrial malfunction and oxidative stress [135].

In the context of chronic liver disease, mitochondria-derived ROS cause hepatocellular injury, inflammation, and fibrosis, thereby intensifying the progression of liver pathology. Understanding the intricate mechanisms through which mitochondria contribute to the production of ROS and oxidative stress is crucial for the development of strategies for countering mitochondrial dysfunction, restoring redox balance, and alleviating oxidative-stress-related damage in disease conditions [2,136].

### 4.2. The Importance of Mitochondrial Metabolism in the Progression of Chronic Diseases

In the liver, mitochondria play a pivotal role both in metabolic functions and in disease pathology. Chronic conditions such as NAFLD, ALD, and liver fibrosis are characterized by mitochondrial dysfunction, oxidative stress, and weakened antioxidant defense mechanisms [137].

Mitochondrial dysfunction arises from various adversities like inflammation and hepatotoxic substances. The liver, a key metabolic organ, requires substantial energy to maintain its crucial functions, especially while experiencing the strain of a chronic disease. Mitochondria meet this demand by producing ATP through oxidative phosphorylation, a highly efficient process. This ATP is fundamental for various liver functions. For instance, the syntheses of proteins and lipids, which are energy-intensive processes, rely heavily on ATP. Additionally, the liver’s role in detoxification, a process that neutralizes and removes toxins and waste, is also ATP-dependent. Any disruption in mitochondrial ATP production can impede the liver’s detoxification capabilities, disrupt the synthesis of essential molecules, and affect ionic balance. Thus, ensuring mitochondrial health and adequate ATP production is vital for managing and potentially mitigating the impacts of liver diseases [138].

Beyond their role in energy production, the mitochondria in liver cells carry out a crucial function in gluconeogenesis. This process involves the conversion of non-carbohydrate substrates, such as lactate derived from anaerobic glycolysis, amino acids from protein catabolism, and glycerol from fat breakdown, into glucose [139]. Mitochondrial enzymes play a pivotal role in this pathway. Enzymes like PEPCK (phosphoenolpyruvate carboxykinase) and G6Pase (glucose-6-phosphatase) are instrumental in the final stages of this pathway, converting oxaloacetate to glucose which can then be released into the bloodstream to maintain blood glucose levels [140]. In the context of chronic liver diseases, this pathway becomes even more significant, particularly during prolonged fasting or under conditions in which the glucose supply is limited [141].

Moreover, the role of mitochondria in fatty acid oxidation holds immense significance in liver metabolism. During this process, fatty acids are broken down, leading to the production of acetyl-CoA. This molecule subsequently enters the citric acid cycle to produce ATP. In the context of chronic liver diseases like NAFLD or alcoholic liver disease, there might be an accumulation of lipids in the liver [142]. Here, efficient mitochondrial fatty acid oxidation becomes imperative to prevent excessive lipid buildup. A malfunction in this process can contribute to the progression of fatty liver diseases as accumulated lipids can cause liver inflammation, fibrosis, and even cirrhosis in prolonged cases. For patients with chronic liver disease, the optimal function of mitochondria in the liver is paramount. They support energy production, regulate glucose and lipid metabolism, assist in the synthesis of ketone bodies and heme, and participate in detoxification pathways [143]. Disruptions in these mitochondrial functions could worsen conditions like NAFLD and metabolic syndrome. Recognizing the central role of mitochondria in liver metabolism is crucial for the effective management and treatment of chronic liver diseases [144].

### 4.3. NRF2 Mediates Mitophagy and Mitochondrial Turnover

NRF2 is a crucial factor in maintaining cellular homeostasis. Its activation can lead to increases in mitophagy and mitochondrial turnover, which are important processes for maintaining mitochondrial quality control [145,146].

Mitophagy is a selective form of autophagy that involves the removal and degradation of damaged or dysfunctional mitochondria. It is a critical process for maintaining a healthy mitochondrial population within cells. Dysfunctional mitochondria can escalate the production of ROS, intensifying oxidative stress, which is a documented antagonist in the progression of CLD [147]. By mediating the removal of dysfunctional mitochondria, mitophagy acts as a defense mechanism, thereby potentially alleviating liver damage. In the context of CLD, this is of paramount importance given the role oxidative stress plays in the disease’s progression [148].

The activation of NRF2 is empirically linked to the enhancement of mitophagy in liver cells. It achieves this by promoting the expression of genes fundamental to both autophagy and mitophagy. Genes like p62/SQSTM1 and LC3, which are central to the process of recognizing and isolating damaged mitochondria during mitophagy, are directly influenced by NRF2. By bolstering these mechanisms, NRF2 reinforces liver health and resilience against CLD [149].

In addition, NRF2 plays a broader role in augmenting mitochondrial turnover, which encompasses the creation and degradation of mitochondria [150]. Key genes involved in mitochondrial biogenesis, such as peroxisome proliferator-activated receptor gamma coactivator 1-alpha (PGC-1α) and mitochondrial transcription factor A (TFAM), witness an uptick in their expression due to NRF2 activation [151]. This, in turn, boosts the replication and synthesis of mitochondrial DNA (mtDNA), fostering the growth of new, healthy mitochondria in liver cells [152].

Healthy mitochondria, in addition playing a role in combating ROS, have another significant function: they contribute to reducing free fatty acids (FFA) and bolster insulin sensitivity. A build-up of FFAs can lead to lipid toxicity, further exacerbating liver damage, while improved insulin sensitivity is critical for glucose metabolism. Both these functions are vital in the context of CLD, with lipid accumulation and insulin resistance among the hallmarks of conditions like NAFLD [153]. Thus, NRF2’s role in promoting healthy mitochondrial populations indirectly supports these two pivotal functions, further underscoring its therapeutic potential in preventing and treating CLD [154,155].

### 4.4. The Interplay of Mitochondria and NRF2 in CLD

In protecting the liver from oxidative stress, mitochondria, in turn, influence the regulatory dynamics of NRF2.

One way mitochondria achieve this is through an enzyme called NAMPT (nicotinamide phosphoribosyl transferase), which is an enzyme that is important to mitochondrial function and instrumental in the biosynthesis of NAD+ (nicotinamide adenine dinucleotide) [156]. NAD+ is vital in cellular redox reactions and bioenergetics, processes which are quintessential for liver cells, especially when dealing with problems like NAFLD or alcoholic liver disease [157]. With reduced mitochondrial health, which is often seen in liver problems due to factors like inflammation, harmful substances, and viral infections, there is a clear change in NAMPT activity [158]. This change results in a drop in NAD+ levels. Later effects impact the SIRT1 pathway, a known controller of NRF2 in liver cells. SIRT1, depending on NAD+, works with NRF2, affecting its defense roles [159].

This relationship shows a two-way connection where in which NRF2 helps keep mitochondria healthy. At the same time, mitochondrial health, guided by the NAMPT-NAD+-SIRT1 path, influences NRF2 activity [156,160].

## 5. Antioxidant Drugs (Clinical Trials) for the Treatment of CLD 

### 5.1. Targeting the KEAP1-NRF2 Complex as a Therapeutic Strategy in Liver Diseases

Antioxidant therapy targeting the NRF2 pathway focuses on its activation to enhance the antioxidant defense system and protect the liver from oxidative-stress-induced damage. Liver diseases, such as NAFLD, ALD, and liver cirrhosis, share a common underlying factor: oxidative stress [161,162]. This strategy shows great promise as a potential management approach for various liver diseases [163]. 

Under normal conditions, KEAP1 binds to NRF2 and targets it for degradation, thereby regulating NRF2 activity. However, certain drugs inhibit KEAP1, leading to the activation and accumulation of NRF2 in the nuclei of liver cells [164,165] (Table 1). Given the impactful role of KEAP1 in modulating NRF2 activity, modifying KEAP1 has emerged as a promising frontier in drug discovery. This is especially pertinent in the context of developing therapeutic interventions designed to ameliorate oxidative-stress-related liver ailments by leveraging the NRF2 pathway [166]. The manipulation of KEAP1 emerges as a novel and promising strategy for the creation of drugs intended to treat various liver diseases by strengthening the body’s inherent antioxidant defenses, a prospect further substantiated via ongoing pre-clinical and clinical studies that reveal the promising potential of drugs targeting KEAP1 [167,168].

### 5.2. Efficacy of Keap1-NRF2-Targeting Therapeutic Agents in Liver Disease Treatment

Among the drugs targeting the Keap1-NRF2 complex, ursodeoxycholic acid (Ursodiol/UDCA) has demonstrated noteworthy therapeutic benefits in patients with cirrhosis. One of the most significant markers of its efficacy is the positive response observed in serum enzyme levels. After UDCA treatment, there was a notable improvement in serum AST, ALT, ALP, GGT, and IgM levels [169]. Additionally, the condition of lobular necroinflammatory lesions, a characteristic feature of cirrhosis, showed marked amelioration. Beyond these physiological indicators, the effect of UDCA is further accentuated at the cellular level. The hepatic expression of 8-OHdG, an oxidative stress marker, was reduced post treatment, suggesting decreased oxidative damage in the liver. Moreover, the hepatic expression levels of both total and phosphorylated Nrf2, key proteins involved in antioxidative responses, increased significantly with the administration of UDCA [169]. This is further supported by elevated levels of antioxidant proteins TRX and TrxR1 in the liver post UDCA treatment, emphasizing the drug’s role in bolstering the liver’s inherent antioxidative defenses. Furthermore, a post-treatment increase in serum TRX levels reiterates UDCA’s systemic antioxidative effect. UDCA elevates cellular defenses against oxidative stress in its efficacy in attenuating cirrhosis, underscoring its potential as an effective therapeutic agent for cirrhosis patients [170].

Oltipraz has also been studied for its ability to activate the NRF2 pathway in the liver. Like bardoxolone and Omaveloxolone, oltipraz enhances the expression of antioxidant and phase II detoxification enzymes [171]. Oltipraz has demonstrated significant therapeutic potential in human trials for NAFLD and liver cirrhosis. Subjects treated with oltipraz, especially those on a high-dose regimen, experienced substantial improvements. The liver fat content was noticeably reduced, insulin resistance was ameliorated, lipid profiles were enhanced, and proinflammatory cytokines decreased. Moreover, ALT and AST levels, pivotal indicators of liver health, showed marked positive changes [172]. In the context of liver cirrhosis, patients receiving both low and high doses of oltipraz displayed improved outcomes in their Ishak fibrosis scores and modified Knodell’s HAI scores. The hepatic collagen area, a marker of fibrosis, also exhibited a declining trend, shedding light on the potential mechanism of oltipraz’s efficacy [173].

Parallel in vivo studies in mice underscored oltipraz’s profound impact on high-fat diet (HFD)-induced changes. Oltipraz was found to boost insulin sensitivity, regulate body weight, reduce fat accumulation, modulate leptin levels, and improve factors such as glucose tolerance, liver gluconeogenesis, adipocyte size, and lipid profile. Central to these effects is oltipraz’s role in activating the endogenous NRF2 system, which seems pivotal in its mechanism of action against detrimental impacts on liver health [171]. Although the findings are promising, it is imperative to note some reported side effects, particularly gastrointestinal disturbances and hepatic function irregularities, in the human trials [172,173]. However, these side effects were limited in comparison to the promising therapeutic outcomes observed in the primary study domains.

Resveratrol (RSV), recognized for its anti-inflammatory attributes and SIRT1 activation, has emerged as a potential treatment for NAFLD. While a meta-analysis indicated marginal enhancements in NAFLD with RSV, distinct clinical markers such as ALT, AST, IL-6, TNF-α, and lipid profiles were employed to assess its effects [174]. Notably, shorter treatments with smaller doses of RSV, like 300 mg daily for three months, manifested beneficial results, including reduced ALT and AST levels, better lipid metabolism, and diminished inflammation [175], yet the prolonged usage of higher doses in some studies did not reflect such benefits [176]; one study even depicted increased levels of ALT and AST [177]. When complemented with lifestyle changes like diet and exercise, RSV’s efficacy in treating NAFLD was augmented [178]. In other research, diverse results emerged. For instance, Timmers et al. found that obese men, when administered 150 mg/day of RSV for 30 days, showed metabolic improvements [179], whereas Poulsen et al. discerned no significant benefits with 500 mg/day of RSV for 4 weeks [180].

In the realm of HCC, RSV suppresses cell viability and inhibits invasive properties in a dose- and time-dependent manner. It induces autophagy, marked by elevated expression of Beclin1 and LC3 II/I, highlighting its role in cellular self-cleansing and its robust antitumor effect. RSV’s modulation of the p53 and PI3K/Akt pathways is pivotal in triggering autophagy and elucidates its anticancer mechanisms, counteracted by 3-MA, Pifithrin-α, and IGF-1 [181].

This multifaceted impact of RSV highlights its potential as a versatile agent in mitigating liver conditions and emphasizes the importance of further exploration into its optimal dosages and therapeutic approaches.

Curcumin, a natural compound, has demonstrated significant efficacy in a range of medical conditions. It has been observed to reduce lipid deposition in models of liver damage by modulating various cellular signaling pathways, such as ERK-p38-MAPK and hepatic Keap1/Nrf2 signaling. Of note is its ability to promote DNA demethylation and inhibit histone deacetylases, which play roles in suppressing the development of HCC [182]. Clinical evidence further substantiates the efficacy of curcumin at treating conditions like HCC, ALD, and especially NAFLD. For instance, in a study conducted by Panahi et al., the administration of phytosomal curcumin at a dose of 1000 mg/day over 8 weeks showed marked reductions in body mass index, and liver enzymes, and displayed improvements in liver sonography [183]. Rahmani et al.’s research echoed similar results in which a lower dose of 500 mg/day resulted in significant improvements in several metabolic and liver-related markers, including reductions in LDL-C, total cholesterol, and elevated HDL-C [184]. 

In the context of ALD, notably, curcumin resulted in 27% and 15.9% reductions in TG and TC levels, respectively, compared to the EtOH group. Additionally, it mitigated oxidative stress in the liver from chronic ethanol exposure by reducing levels of ROS and MDA, maintaining hepatic GSH levels, and preserving GPx and GST activities, which are essential for antioxidative defense. This highlights curcumin’s potential as a treatment for ethanol-induced liver abnormalities and oxidative stress [185].

Importantly, despite its known challenges with bioavailability, both clinical and preclinical studies have consistently confirmed curcumin’s protective and therapeutic effects in oxidative-associated diseases, including liver disorders. It is posited that the underlying therapeutic activities of curcumin might predominantly arise from its main metabolites, which seem to play a pivotal role in its biological function.

The therapeutic potential of Liraglutide, previously studied in high-fat diet (HFD) mice, was highlighted in attenuating NAFLD via the modulation of NRF2. In an in vivo study, Liraglutide reduced body weight and fat mass, improved dyslipidemia, and ameliorated glucose homeostasis. Notably, the drug enhanced NRF2 expression in the liver, leading to the upregulation of crucial downstream genes, including Catalase, NQO1, and GCLM, highlighting its pivotal role in combating NAFLD [186].

Transitioning from mice to clinical settings, a study on NAFLD patients revealed Liraglutide’s pronounced effects. Patients reported decreased HbA1c levels, body weight, BMI, and altered fat distribution, leading to diminished visceral and subcutaneous fat areas. Alongside these, significant biochemical alterations such as reduced ALT, GGT, and plasma triglyceride levels and increased HDL-C and plasma adiponectin levels were observed. The considerable reduction in the liver fat content (LFC) was particularly salient [187]. A subsequent meta-analysis further vouched for Liraglutide’s efficacy, highlighting mild declines in liver enzymes and its positive impact on BMI, lipoproteins (both HDL and LDL), HbA1c, and triglycerides [188].

However, while Liraglutide promises therapeutic benefits, its side effects warrant attention. Many patients experienced gastrointestinal reactions typical of GLP-1 receptor agonists like Liraglutide. Symptoms ranged from mild, such as nausea and indigestion, to severe cases of vomiting or diarrhea. It is paramount for patients and healthcare providers to meticulously monitor these reactions, ensuring an informed balance between the drug’s benefits and potential risks [188].

Bardoxolone methyl (BM) is under scrutiny for its ambiguous effects in treating chronic liver diseases as an NRF2 activator [189]. Various studies illustrate its ability to upregulate antioxidant and detoxifying enzymes by inhibiting KEAP1, leading to the mitigation of inflammation and oxidative stress. BM demonstrates a substantial positive impact on lipid metabolism and insulin sensitivity. It aids in reducing lipid accumulation and steatosis in the liver, particularly under a high-fat diet, and in improving the regulation of blood glucose levels [190]. The subsequent enhanced insulin sensitivity and glucose metabolism are pivotal for managing chronic liver conditions, such as ALD and NAFLD/NASH, and have been highlighted in various in vivo studies and human clinical trials, illustrating BM’s capacity to establish an anti-inflammatory state in hepatic tissues [163]. 

However, the landscape of research presents a dichotomy. While some studies and clinical trials underscored BM’s positive impact on liver functionality and chronic liver disease, contrasting studies revealed alarming effects. These opposing findings suggest that BM may elevate ALT levels, potentially leading to liver damage, a dire contradiction to its therapeutic benefits. The severe side effects observed, including muscle spasms, heart failure, and fatal outcomes, cast further shadows on its prospective approval as a therapeutic agent [191,192].

Given the conflicting evidence, the medical community is pressed to conduct comprehensive and holistic studies to elucidate BM’s precise therapeutic potential and risks and its contradictory role in liver health.

Omaveloxolone (RT-408) is emerging as a notable agent in addressing chronic liver diseases, attributed to its role in upregulating NRF2 expression and displaying antioxidative and anti-inflammatory properties. It shows promise in improving the liver’s architecture by reducing fat deposition and inflammatory cell infiltration in NASH models, indicating potential efficacy in ameliorating CLD. Additionally, the drug reduces collagen and lipid accumulation in the liver and induces hepatoprotective Nrf2 target genes. Furthermore, it aids metabolic management by improving glucose control and altering lipid profiles, enhancing HDL cholesterol levels [193].

However, human clinical trials revealed a blend of beneficial effects and noteworthy adverse events. Patients experienced mild to moderate side effects like headaches, nausea, and elevated ALT and AST levels, primarily within the initial 12 weeks, with a marked decrease in frequency between weeks 12 and 48, emphasizing the need for prolonged, meticulous studies to comprehensively assess Omaveloxolone’s overall therapeutic effect and safety. The reversible nature of adverse events and the absence of hepatic injury symptoms are pivotal observations, indicating potential therapeutic promise [194]. 

Omaveloxolone’s potential in mitigating chronic liver conditions and its blend of positive effects with reversible adverse events demand careful evaluation through long-term, rigorous studies to conclusively validate its efficacy and safety [195]. 

Antioxidant therapy targeting NRF2 activation exhibits substantial promise for treating chronic liver diseases such as NAFLD, ALD, and liver fibrosis by mitigating oxidative-stress-induced damage. The agents that were described show considerable therapeutic benefits, including the amelioration of inflammation, liver enzyme normalization, and improved lipid metabolism [196]. However, the efficacy of these agents is nuanced, with varying results on liver health parameters and a range of side effects, highlighting the need for personalized therapeutic approaches and optimal dosing strategies. Notably, several agents displayed adverse effects in short-term treatments, necessitating long-term and comprehensive studies to delineate the full spectra of their therapeutic windows, safety, and holistic impacts. Interestingly, some studies demonstrated that the drug exhibited hepatoprotective effects even in the absence of NRF2, highlighting its multifaceted mechanisms of action [197]. Nonetheless, the contrasting outcomes and serious adverse effects observed, particularly with Bardoxolone methyl, underscore the urgent need for the meticulous evaluation and validation of these agents’ therapeutic potentials and risk profiles. Consequently, while the initial findings are promising, the convergence of enhanced therapeutic outcomes with minimized adverse events requires rigorous, in-depth exploration, keeping patient safety at the forefront [82,165,198,199]. 

## 6. Challenges and Limitations: The Double-Edged Role of NRF2 in Chronic Liver Diseases

In the intricate realm of chronic liver diseases, NRF2 has a multifaceted role that stretches beyond its well-established cytoprotective and antioxidant capacities. Although NRF2 is fundamental to cellular defense, particularly in combating liver diseases, its wider roles in NAFLD, ALD, and HCC present a complex landscape which is intricately tied to inflammation, fibrosis, and malignancy. Nrf2’s role in these chronic conditions may be double-edged [200].

Inflammation remains a critical phase in the progression of NAFLD and ALD and is a prevalent precursor to HCC. Acutely activated NRF2 bolsters the liver’s defense against oxidative insults, exemplified via its upregulation of genes like ADH and ALDH, which combat alcohol-induced oxidative stress. However, the darker side of this mechanism in chronic alcohol consumption, as seen in ALD, is the resultant buildup of acetaldehyde, a toxic metabolite that triggers inflammatory processes by activating Kupffer cells and drawing neutrophils into the liver [201]. In the NAFLD scenario, NRF2’s sway over lipid metabolism is pronounced. Its prolonged stimulation of lipogenic genes indirectly encourages hepatic fat buildup and a subsequent pro-inflammatory state marked by the secretion of cytokines like TNF-α and IL-6, further exacerbating liver damage [89].

In the domain of malignancies, particularly HCC, NRF2’s role becomes even more intricate. Its continuous activation in liver diseases can be a harbinger of cancer progression [202]. Intriguingly, the presence of Mallory–Denk bodies (MDBs) and intracellular hyaline bodies (IHBs), characteristic of specific HCC subtypes, is interlaced with NRF2’s function. The main component of IHBs, p62, correlates with shorter survival in HCC patients, and its accumulation is synonymous with dysfunctional autophagy and enduring NRF2 activation [203]. This interrelation has repercussions in the progression of HCC, with heightened glutathione (GSH) production potentially causing chemoresistance and amplifying the proliferative potential of hepatoma cells [204]. Further emphasizing NRF2’s influence, its overexpression has been observed to modulate apoptosis, increasing anti-apoptotic factors like Bcl-xL while diminishing pro-apoptotic entities like Bax, thus providing a survival boon to malignant cells [205].

NRF2 indisputably occupies a pivotal position in hepatoprotective and antioxidant mechanisms, offering potential therapeutic avenues in the treatment of liver diseases. Its capacity to bolster the liver’s defenses against oxidative threats is a testament to its foundational role in cellular defense. Additionally, the prolonged activation of NRF2 has been shown to lead to increased lipid accumulation in the liver due to the upregulation of genes associated with lipid synthesis and storage [206]. As our understanding deepens, it becomes evident that the benefits of NRF2 are intertwined with potential pitfalls. The prolonged activation of NRF2, while protective in certain scenarios, can paradoxically exacerbate the progression of specific liver diseases. As a result, therapeutic interventions harnessing NRF2 activators should be judiciously considered and tailored to the unique conditions of each patient. Furthermore, any therapeutic approach must concurrently evaluate other influential factors, ensuring that the multifaceted role of NRF2 is both respected and harnessed optimally for the betterment of patient outcomes.

## 7. Concluding Remarks and Perspectives

In conclusion, oxidative stress plays a significant role in the pathogeneses and progression of CLDs. This stress orchestrates a cascade of detrimental events: hepatocellular damage, inflammation, fibrosis, and the activation of hepatic stellate cells, leading to liver fibrosis. The interplay between oxidative stress and inflammatory pathways constitutes a relentless cycle, exacerbating liver injuries and accelerating disease progression.

In the labyrinth of liver metabolism, mitochondria stand as irreplaceable powerhouses, underpinning ATP generation and orchestrating pivotal hepatocyte operations. They catalyze processes such as gluconeogenesis, fatty acid oxidation, and ketone body synthesis, particularly during carbohydrate shortfalls. Additionally, they play a vital role in heme synthesis, which is indispensable for oxygen transport and metabolism. However, these critical functions, especially their detoxification role, can be jeopardized by toxins, casting shadows on liver health. Chronic conditions like NAFLD, ALD, and liver fibrosis spotlight the susceptibility of mitochondria to dysfunctions induced by inflammatory cytokines, hepatotoxins, and metabolic aberrations.

Amidst this complexity, NRF2 surfaces as a beacon of defense. As a transcription factor, it shoulders the responsibility of shielding cells from oxidative stress and upholding cellular equilibrium. NRF2 not only orchestrates the antioxidant gene response to battle ROS but also fortifies mechanisms like mitophagy, which is pivotal for mitochondrial integrity. Its activation champions the elimination of malfunctioning mitochondria, bolstered by genes like p62/SQSTM1 and LC3, and stimulates new mitochondrial generation through genes such as PGC-1α and TFAM. The enzyme NAMPT, ensconced within mitochondria, augments NRF2 activation through the NAD+-SIRT1 axis. This multifaceted relationship emphasizes NRF2’s therapeutic potential, highlighting pathways which guard mitochondrial structure and shield cells from oxidative duress.

Integrating this with our earlier understanding of oxidative stress in CLD, it is evident that combating this menace extends beyond the mere management of ROS. The intricate dance between mitochondrial function, NRF2, and oxidative stress provides a rich tapestry of potential therapeutic interventions. Continuing to unravel these complexities is not only vital for pioneering treatment strategies but also holds immense promise for improving the life quality of those battling CLD. Through persistent research and the exploration of these molecular intricacies, we edge ever closer to transformative strategies that can redefine the prognosis and therapeutic landscape of CLD.

## Figures and Tables

**Figure 1 antioxidants-12-01928-f001:**
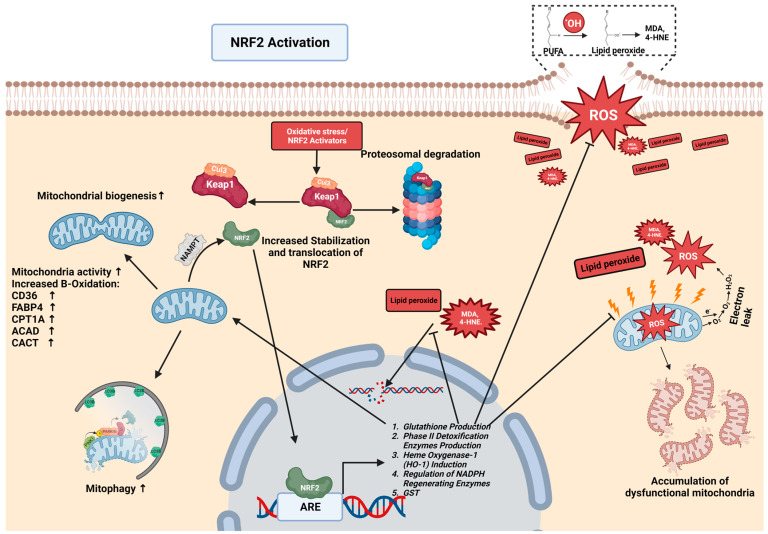
NRF2 regulation and NRF2-mitochondrial interplay in chronic liver disease. In chronic liver disease, elevated levels of ROS lead to the oxidation of membranes, particularly polyunsaturated fatty acids (PUFAs), culminating in lipid peroxidation. Subsequently, lipid peroxides are converted into aldehydes, such as malondialdehyde (MDA) and 4-hydroxynonenal (4-HNE), which further inflict damage on mitochondria and DNA. This cycle results in an accumulation of dysfunctional mitochondria and DNA mutation. Normally, NRF2 is complexed with Keap1-Cul3, targeting it for proteasomal degradation. NRF2 activators (oxidative stress/NRF2-activator drugs) modify KEAP1, stabilizing NRF2 and facilitating its translocation to the nucleus. In the nucleus, NRF2 binds to the antioxidant response element (ARE), initiating the transcription of proteins that reduce the levels of ROS, inhibit lipid peroxidation, neutralize lipid peroxides and aldehydes, and bolster mitochondrial biogenesis, activity, and turnover. This cascade also enhances the expression of proteins like CD36, CPT1A, ACAD, and CACT, promoting beta-oxidation and mitophagy. Mitochondria produce the protein NAMPT, which further activates NRF2, establishing a feedback loop between NRF2 and mitochondrial function. All graphical figures are created by using BioRender (https://biorender.com; accessed on 27 September 2023).

**Figure 2 antioxidants-12-01928-f002:**
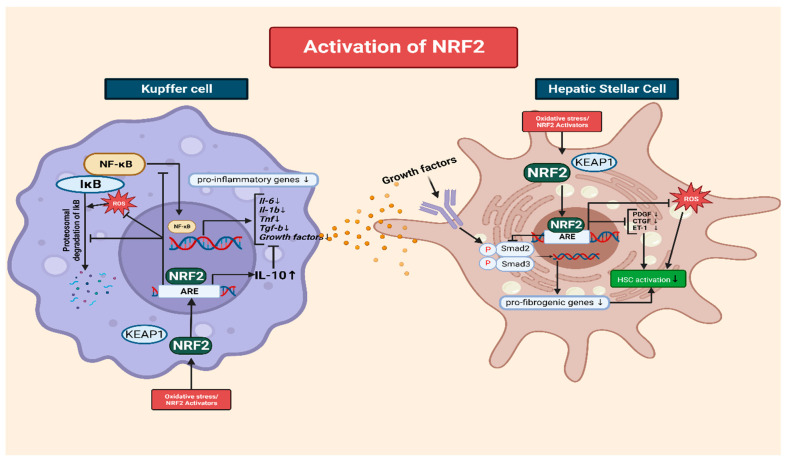
The interplay of NRF2 in Kupffer cells and HSCs during the progression of chronic liver disease. In the context of CLD, elevated levels of ROS play a pivotal role in amplifying inflammation. An increase in ROS accelerates the degradation of IκB, a natural inhibitor of NF-κB. Therefore, more NF-κB translocates to the nucleus, fostering the production of pro-inflammatory and growth factors such as IL-1B, IL-6, and TNF-a. Kupffer cells then secrete these cytokines, among which growth factors are recognized by hepatic stellate cells (HSC) via receptors. This recognition process subsequently triggers the formation of SMAD complexes that migrate to the nucleus, promoting fibrotic responses. NRF2 has a dual role in both inhibiting the pro-inflammatory NF-κB signaling pathway and modulating HSC activation. NRF2 prevents the degradation of IκB and curtails the translocation of NF-κB. This transcription factor further dampens the inflammatory landscape by inhibiting the release of pro-inflammatory cytokines and bolstering the formation of IL-10, an anti-inflammatory cytokine. Specifically, within HSCs, NRF2 thwarts the formation of fibrosis-promoting SMAD complexes and diminishes the expression of fibrogenic mediators like PDGF, CTGF, and ET-1. All graphical figures are created by using BioRender (https://biorender.com; accessed on 27 September 2023).

**Table 1 antioxidants-12-01928-t001:** Compilation of therapeutic compounds targeting the NRF2-Keap1 pathway in CLD.

Compound	Type	Mechanism of Action	Clinical Trial	Registration Number
Bardoxolone	Synthetic Triterpenoid	Keap1 Modification	Phase I	NCT01563562
Omaveloxolone (RTA-408)	Synthetic Triterpenoid	Keap1 Modification	Phase I	NCT03902002
Oltipraz	Synthetic Dithiolethione	Keap1 Modification	Phase II	NCT00956098
Liraglutide	Synthetic Peptide	Keap1 Modification	Phase II	NCT01237119
Ursodiol	Bile Acid	Keap1 Modification	Phase IV	NCT05849558
Resveratrol	Nonflavonoid Polyphenol	Keap1 Modification	Phase II	NCT02216552
Curcumin	Diarylheptanoid	Keap1 Modification	Phase II	NCT04109742

## Data Availability

No new data were created or analyzed in this study. Data sharing is not applicable to this article.

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
