# Peer review of "The Roles of NFR2-Regulated Oxidative Stress and Mitochondrial Quality Control in Chronic Liver Diseases"

_antioxidants, 2023, doi:10.3390/antiox12111928_

Round 1

Reviewer 1 Report (Previous Reviewer 1)

Comments and Suggestions for Authors

The authors have made an extensive work of editing and the manuscript is now consistent in form and substance.

Author Response

Dear Reviewer,
Please find attached the file. Your expertise and feedback was invaluable in enhancing the quality of our work.

Sincerely yours,
Yoon Seok Roh D.V.M., Ph.D.
College of Pharmacy and Medical Research Center
Chungbuk National University
194-21 Osong-Saeng-Myeong 1-Ro, Cheong-Ju 28160, South Korea Phone
number: +82-43-261-2819
Email address: ysroh@cbnu.ac.kr

Reviewer 2 Report (Previous Reviewer 2)

Comments and Suggestions for Authors

While the revised manuscript shows marked improvement, several critical issues have yet to be adequately addressed. For instance:

1. The author frequently employs ambiguous descriptions. For instance, in "section 3.2: By modulating these genes, NRF2 regulates the cellular uptake of fatty acids and lipoproteins, which is vital for preventing excessive lipid accumulation in the hepatocytes." The author does not specify whether NRF2 positively or negatively modulates lipid uptake. Based on the references provided, NRF2 promotes lipid uptake. Does its role in CLD align with the protective/resistant roles described in other sections? This discrepancy is precisely what should be addressed in this review article.

2. Section 3.3 consists of seven paragraphs, yet only the opening sentence and the final paragraph, which is a mere four lines, address NRF2.

3. In section 3.4, how many of the cited references actually explore the interaction between NRF2 and NF-κB in Kupffer cells?

4. The title of reference 161 is 'Oltipraz ameliorates the progression of steatohepatitis in Nrf2-null mice fed a high-fat diet.' This suggests that oltipraz's therapeutic effects against NAFLD is independent of NRF2, given its efficacy in NRF2-null mice. This perspective contrasts sharply with the main theme of section 5. Furthermore, the relevant description associated with reference 161 does not address this discrepancy. So, of the compounds/drugs mentioned in this review, how many have hepatoprotective effects that are reliant on the NRF2/Keap1 pathway?

Author Response

Dear Reviewer,
Please find the attached the file. Your expertise and feedback will be invaluable in enhancing the quality of our work.

Sincerely yours,
Yoon Seok Roh D.V.M., Ph.D.
College of Pharmacy and Medical Research Center
Chungbuk National University
194-21 Osong-Saeng-Myeong 1-Ro, Cheong-Ju 28160, South Korea Phone
number: +82-43-261-2819
Email address: ysroh@cbnu.ac.kr

Round 2

Reviewer 2 Report (Previous Reviewer 2)

Comments and Suggestions for Authors

The manuscript has been significantly improved and is now acceptable in its current form for publication. Congratulation!

This manuscript is a resubmission of an earlier submission. The following is a list of the peer review reports and author responses from that submission.

Round 1

Reviewer 1 Report

Comments and Suggestions for Authors

The Park, Rustamov and Roh manuscript is an interesting article that reviews the key phenomenon of oxidative stress in chronic liver diseases, especially in the so prevalent non-alcoholic and alcoholic etiologies. Although this subject has been profusely treated previously, they stress the important role of NRF2 and mitochondria in the regulation of oxidative stress and update the published data. Finally, authors point out NRF2 as an appropriate therapeutic target and review a few new strategies that could be beneficial for chronic liver diseases through the modulation of NRF2 and mitochondrial pathways. The English is very good and the figures are nice.

Comments:

Major.

- The manuscript is structurally uneven. From my point of view, this should be fixed. Section 2 is divided in sections and subsections, sometimes very short ones. In contrast point 3 and 4 are continuous longer sections. Please unify the style.

- There are a few concepts that are repeated too many times. Please re-read the whole manuscript and try to eliminate unnecessary repeated information. This overload the text and extend for no reason the length of the manuscript. 

Some examples are (not the only ones):

Line 83: Oxidative stress is defined as an imbalance between the production of ROS and the antioxidant defense system in the body, which leads to various inflammatory diseases, CLDs, and cancer [3,28,29].

Line85:  It is a result of excessive ROS accumulation or decreased capacity of antioxidant defense mechanisms to effectively neutralize them. Please read the whole manuscript and try to eliminate unnecessary repetitions

Line 196: In CLD, an imbalance exists between free radical and ROS production and antioxidant defense systems, leading to increased oxidative stress.

Also, in Line 289 (Dysregulation of lipid metabolism plays a central role in CLDs such as NAFLD and NASH. An imbalance between lipid uptake, synthesis, and degradation leads to lipid accumulation within hepatocytes, contributing to hepatosteatosis, inflammation, and disease progression) there is no need to repeat this again; there is a great definition of the dysregulation of lipid metabolism in the progression of NAFLD in line 44. 

- In line 83 also, there is a general definition of oxidative stress, not specific to liver diseases, so I would change CLD by organ dysfunction.

- In section 2.2 I would say Formation of ROS, because Free radicals is a typo of ROS and authors are talking about free radicals but also about non-radical species.

- Title of section 4 is very promising and the one that everyone wants to know about. However, the content remains very superficial and needs to be well developed, instead of being a list of 3 drugs that apparently have the same mode of action and no data of their effects in patients with liver diseases.

Minor:

- In the abstract sections, I would eliminate “through genes like p62/SQSTM1, LC3, PGC-1α, and TFAM”. It is too specific and the authors don’t extend the information about   these genes in the manuscript. Moreover, gene names are in the abbreviated form and, if mentioned, they should be explained.

- Line 367 is repeated twice: NRF2 regulates inflammation through multiple mechanisms. NRF2 regulates inflammation through multiple mechanisms.

- Line 483: shown should be shows

- Line 483: this sentence seems out of place: Liver diseases, such as NAFLD, ALD, and liver fibrosis, share a common underlying factor: oxidative stress [131,132].

Reviewer 2 Report

Comments and Suggestions for Authors

1.       While there are debates about NRF2's roles in NAFLD, the authors have not addressed the contentious aspects of NRF2's involvement in the disease.

2.       The authors highlight that NRF2 plays a growing part in mitochondrial function, governing mitochondrial biogenesis and quality control. They emphasize that activating NRF2 can enhance mitochondrial function and diminish oxidative stress in multiple cell types, inclusive of liver cells. However, they do not delve into NRF2's specific impact on mitochondria in the context of NAFLD or CLD.

3.       The review spans 15 pages, excluding the references, yet the discussion dedicated to NRF2's influence on mitochondria is confined to merely one paragraph, constituting a minute portion of the entire article.

4.        What triggers NRF2 activation as shown in Figure 1? While Figure 2 illustrates the role of the Kupffer cell, it lacks a corresponding description in the main content. It would be beneficial to distinctly elaborate on the functions of NRF2 in Kupffer cells and/or HSCs, especially concerning the progression of inflammation and fibrosis.

5.       Is a Keap1 modifier a viable option for drug development aimed at NRF2 activation? It is imperative to consider this in this article. While Bardoxolone is known for its adverse effects, what about the potential side effects linked with bardoxolone and RTA-408, a bardoxolone derivative? As for Liraglutide, ursodiol, resveratrol, curcumin, and sulforaphane, is their hepatoprotective effect reliant on the NRF2/Keap1 pathway? If not, their inclusion in this paper might be debatable.